# Episodic-like memory trace in awake replay of hippocampal place cell activity sequences

**Susumu Takahashi***

Laboratory of neural circuitry, Graduate School of Brain Science, Doshisha University, Kyoto, Japan

**Abstract** Episodic memory retrieval of events at a specific place and time is effective for future planning. Sequential reactivation of the hippocampal place cells along familiar paths while the animal pauses is well suited to such a memory retrieval process. It is, however, unknown whether this awake replay represents events occurring along the path. Using a subtask switching protocol in which the animal experienced three subtasks as 'what' information in a maze, I here show that the replay represents a trial type, consisting of path and subtask, in terms of neuronal firing timings and rates. The actual trial type to be rewarded could only be reliably predicted from replays that occurred at the decision point. This trial-type representation implies that not only 'where and when' but also 'what' information is contained in the replay. This result supports the view that awake replay is an episodic-like memory retrieval process.

## Introduction

One of the most important aspects of episodic memory is the mental retrieval of personal recollections of what happened "where" and "when", referred to as "mental time travel" (*Tulving, 1983*). Episodic memory retrieval is important not only for remembering recent events but also for imagining future ones (*Hassabis et al., 2007*). The hippocampus is critically involved in episodic memory retrieval, and within this structure reside place cells—principal cells exhibiting place-specific firing patterns in particular locations. The presence of such cells has led to the hypothesis that the hippocampus is the locus of the "where" element of episodic memory (*O'Keefe and Dostrovsky, 1971*). Following on from this seminal finding, more recent studies have demonstrated that hippocampal place cell activity simultaneously represents a broad range of aspects of memory, including odors and their match/non-match status in a particular location, called the "place field", as well as elapsed time (*Otto and Eichenbaum, 1992*; *Sakurai, 1996*; *Wood et al., 1999*; *Pastalkova et al., 2008*; *MacDonald et al., 2011*). In the place field, both firing rate and location can change independently and are affected by external factors, such as spatial environment and task demand, and internal factors such as prospective and retrospective memory (*Markus et al., 1995*; *Frank et al., 2000*; *Wood et al., 2000*; *Anderson and Jeffery, 2003*; *Ferbinteanu and Shapiro, 2003*; *Smith and Mizumori, 2006*). During maze running, place cell activity sequences in rodents convey information on both familiar paths and accompanying subtasks (*Allen et al., 2012*; *Takahashi, 2013*).

Within sharp wave/ripples (SWRs) in local field potentials (LFPs) during slow-wave sleep or periods of awake immobility, internally generated place cell activity sequences are often reactivated in a temporally compressed manner (*Wilson and McNaughton, 1994*; *Foster and Wilson, 2006*; *O'Neill et al., 2006*; *Diba and Buzsáki, 2007*; *Karlsson and Frank, 2009*; *Gupta et al., 2010*). During brief periods of immobility, pivotal paths to a remembered goal can be predicted, regardless of the location in which the SWRs have previously occurred (*Pfeiffer and Foster, 2013*). Such "replay" is

*For correspondence: stakahas@ mail.doshisha.ac.jp

**Competing interests:** The author declares that no competing interests exist.

**eLife digest** Place cells are neurons that respond to a particular location in the physical world. For example, as a rat runs around a maze, some place cells will become active when the rat reaches one corner. When the rat moves on towards a different corner, other place cells activate instead. The real-time activity of these place cells helps the rat to work out where it is in the maze. This activity also contains information about what the rat is doing.

In addition to their real-time activity, place cells also help previous events to be 'replayed' mentally, which is important for making decisions. Previous studies have shown that when a rat pauses during a task, place cell replays allow it to mentally map out the route it needs to take. However, it is less clear whether these replays also provide information about what the rat needs to do.

Takahashi gave rats a number of tasks to perform inside a figure of eight maze. In one of these tasks, the rat had to assess which one of two lights was lit up, and run towards it. In the other two tasks, the rat had to remember the direction it took on the previous occasion, and go in the opposite direction. During these tasks, the rat would occasionally pause to replay information about the task.

Takahashi recorded what the rats' place cells were doing during these pauses, and found that the place cell replays contained information about both the path the rat needed to take ('where and when' information), and which task it needed to carry out ('what' information). This suggests that replays are important for the ability to recall information about specific events, which is known as episodic memory. Takahashi's results may therefore also help us to learn more about this "mental time travel" and human conditions that damage episodic memory, such as Alzheimer's disease.

thus considered a neuronal substrate for retrieving memories on familiar paths (*Carr et al., 2011*). However, previous studies investigating the content of replays have primarily focused on the geography and timeline of the animals' running path as experienced in the environment. It is therefore unclear whether the replay conveys information on events (i.e., "what" information) occurring along the reactivated path.

I recently recorded the ensemble activity of place cells in the hippocampal CA1 of rats made to continuously navigate four journeys in a figure-eight maze under three intermittently switching subtasks within a single session: visually guided discrimination (VD), non-delayed spatial alternation (NA), and delayed spatial alternation (DA) (*Takahashi, 2013*). This task design allowed for comparison of hippocampal neuronal activity between and within different journeys and subtasks. I found that while the animal is running in the maze, differences between individual journeys and between subtasks in the maze are independently encoded in the firing locations and rates of place cells, respectively. Furthermore, all trial types (i.e., journey type and subtask) experienced in the maze could be decoded from the ensemble activity of place cells. Since the rats ran along similar spatial paths while performing different subtasks throughout this task, differences between subtasks can be interpreted as non-spatial "what" information. Provided that the mechanisms are preserved in temporally compressed replays that occur during brief periods of immobility, the subtask as "what" information occurring along the path in the maze might be examined in the replay. Building on this previous study (*Takahashi, 2013*), I accordingly investigated whether "what" information is contained in the replay of place cell activity sequences during the brief immobility periods that occur while the animal is engaging in the task.

## Results

According to the subtask switching protocol, four rats were trained for about one hour in a specific sequence—VD (20 laps), NA (20 laps), VD (10 laps), DA (20 laps), VD (20 laps), NA (20 laps), VD (10 laps), and DA (20 laps)—and were rewarded each time they arrived at the reward zone at the correct return rail (see 'Materials and methods'). In the VD subtask, a cue light on the left or right side was illuminated to indicate which direction to turn at a decision point (*Figure 1A*) to receive a reward. In both the NA and DA subtasks, cue lights on both sides were illuminated and the rats had to alternate between left and right at the decision point (*Figure 1B*). The difference between the NA and DA subtasks was that in the DA subtask, a barrier appeared in the middle of the central stem for five

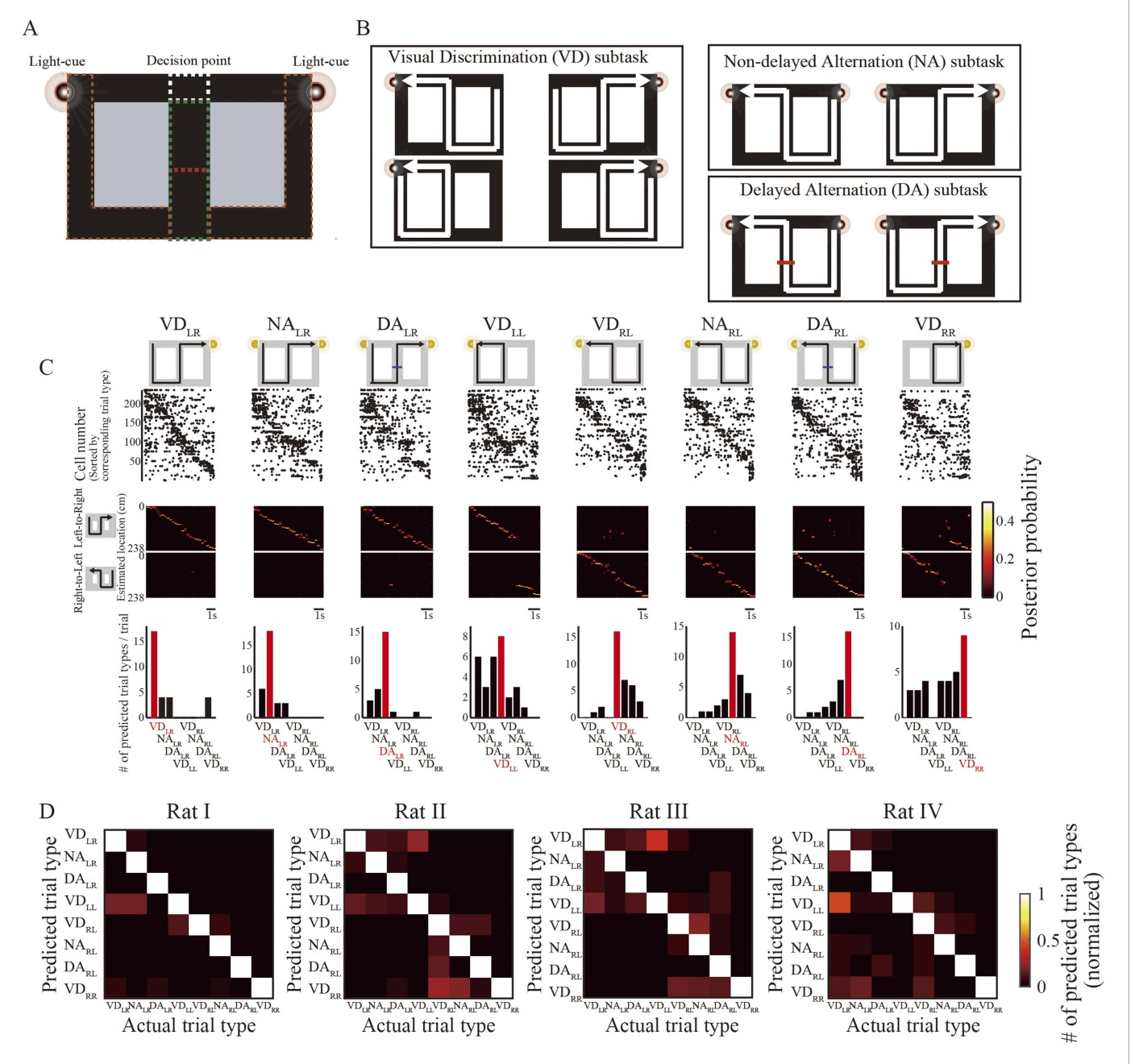

**Figure 1.** Decoding path and trial-type prediction during running. (**A**, **B**) Task design and configuration. (**A**) Each task could only be identified at the decision point (junction; white dotted rectangle), because the rats could not see the visual cues until they reached that point even in the central maze stem (green dotted rectangle). If correct, the rats received electrical stimulation of the medial forebrain bundle (MFB) as a reward, beneath the visual cue. After receiving a reward, the rats always briefly stopped within the start zones (brown dotted rectangles). In the delayed spatial alternation (DA) subtask, the rats had to wait for 5 s in the maze stem, until the barrier wall (red dotted line) disappeared. (**B**) Visual cue configuration and possible journeys during task performance. In the visually guided discrimination (VD) subtask, one of the visual cues was illuminated to guide the rat towards its goal. However, in the non-delayed spatial alternation (NA) and DA subtasks, spatial memory retrieval was required for the rat to choose the opposite goal to the previous one, because both of the visual cues were illuminated. (**C**) Actual trial types are organized in eight columns. The topmost illustration describes actual trial type. Below that, representative raster plots show spiking activity recorded from rat I for each trial type in a single trial. Each dot represents an action potential. The middle graph in each column shows the posterior probability of paths decoded by the Bayesian decoder involving left-to-right (upper) and right-to-left (lower) journeys (values indicated by color bar at right). At the bottom of each column, a bar graph gives the number of predicted trial types by the trial-type prediction method. Note that the most frequently predicted trial type (red bar) matched the actual one (red label) for all trial types. (**D**) Confusion matrices between predicted and actual trial types for each rat during the entire session (values indicated by color bar at right). Note that the prominent

*Figure 1. continued on next page*

*Figure 1. Continued*

diagonal line in each matrix shows that predicted trial type matched the actual one in most cases for all rats.

The following figure supplement is available for figure 1:

**Figure supplement 1**. Spatial pattern of place cell activity.

seconds to impose a delay period. The subtask to be rewarded (i.e., VD, NA, or DA) could therefore only be distinguished at the decision point, based on visual cues (*Takahashi, 2013*).

## Place cell sequences represent path as well as trial type during running

I tested whether the methods used for the subsequent replay analyses could predict running path and trial type from the place cell activity sequences recorded during running. A total of 1084 principal cells in the dorsal hippocampal CA1 were recorded using an array of 10 extracellular dodecatrodes (*Takahashi and Sakurai, 2005*) in four well-trained rats (individuals that had achieved an overall task performance of >90% for a week), while they were performing the task (*Table 1*). Because running speed, head direction, and physical position can influence place cell firing (*McNaughton et al., 1983*; *Wiener et al., 1989*), only 556 place cells were examined in the following analyses. These cells, which had spatial information >0.3 bits/spike, fired at significantly different rates during different trial types (i.e., journey type and subtask), even when running speed, head direction, and x and y coordinates were taken into account (Analysis of covariance (ANCOVA) with these covariates, $p < 0.05$) (*Wood et al., 2000*). The rats were exposed to each of the subtasks intermittently at least twice throughout the entire session, but my previous study suggested that place fields are not remapped between trials of the same type, regardless of differences in levels of exposure to the trials (*Takahashi, 2013*). I therefore combined place cell activity from all trials of the same type, irrespective of the level of exposure.

Differences between individual journeys are encoded in the timing of firing across place cells while the animal is running (*Figure 1—figure supplement 1A,B*) (*Takahashi, 2013*). For each trial type, the rats' paths could be accurately decoded from the place cell activity sequences (*Figure 1C*, top) using a memoryless Bayesian decoder (*Figure 1C*, middle; median error: 6.0–13.5 cm). Since similar paths could be decoded from the place cell activity sequences during trial types with the same journey but different subtasks (e.g., VD$_{LR}$, NA$_{LR}$, and DA$_{LR}$), the Bayesian decoder cannot per se identify trial types. As my previous study reported (*Takahashi, 2013*), differences between subtasks are encoded in firing rates across place cells. Therefore, using a prediction method based on these firing rates (*Allen et al., 2012*), I predicted trial type from the firing rate pattern across place cells in conjunction with the decoded path. The bottom portion of *Figure 1C* shows trial types predicted from rat I in a single trial. In the visually guided VD$_{RR}$ and VD$_{LL}$ trials, some mismatches occurred because the rats could not accurately predict the trial type until they reached the junction. The predictions were most accurate for the spatial alternation trials. Confusion matrices between predicted and actual trial types for each rat during the entire session (*Figure 1D*) show that a similar pattern was observed for all rats (overall mean, 74.3–84.5% (leave-one-out estimate), chance = 12.5%; *Table 1*). The Bayesian decoder and the prediction method together are sufficient for interpreting the representation of path and trial type in place cell activity sequences.

## Rat behavior during brief periods of immobility

Before examining the replays, I investigated the rats' behavior during the brief periods of immobility, when the replays occurred and the animals' behavior suggested that they were actively gathering information (*Benjamini et al., 2011*). Specifically, I assessed the directions in which the rats' heads were pointing at these times. In correct trials, the rats' heads pointed towards a goal in the opposite direction from the respective start zone (Watson–Williams test, *P*: $1.1 \times 10^{-16}$, n = 588 (left), 1305 (right); Rayleigh test; *P*: left, $1.1 \times 10^{-56}$, right, $4.9 \times 10^{-324}$) and stem (Watson–Williams test, *P*: $1.1 \times 10^{-16}$, n = 257 (left), 395 (right); Rayleigh test; *P*: left, $1.7 \times 10^{-235}$, right, $5.2 \times 10^{-125}$) (*Figure 2*). This suggests that the rats sampled a memory-guided goal in preparation for switching to either the NA or the DA subtask. In this experiment, head direction is therefore considered a behavioral sign indicating expected future choice. In erroneous trials, however, head direction was ambiguous only in the maze

**Table 1**. Behavioral and electrophysiological measurements and estimation accuracy of position and trial-type during running

| RAT number | I | II | III | IV | Total |
|---|---|---|---|---|---|
| Number of erroneous laps | 0 | 9 | 1 | 4 | 15 |
| Number of correct laps | 142 | 142 | 139 | 139 | 562 |
| Task performance (%) | 100% | 94.0% | 99.3% | 97.2% | 97.4% |
| Number of principal cells | 412 | 193 | 202 | 277 | 1084 |
| Number of place cells (which meet the defined criteria) | 238 | 97 | 110 | 111 | 556 |
| Number of interneurons | 24 | 15 | 10 | 12 | 61 |
| Unit isolation quality (isolation distance, mean ± SEM) | 45.4 ± 7.9 | 23.3 ± 1.6 | 33.6 ± 2.1 | 20.6 ± 0.9 | |
| Accuracy of position estimation during running (overall median) | 6.0 cm | 6.5 cm | 13.1 cm | 13.5 cm | |
| Accuracy of trial-type prediction during running (overall mean) | 89.5% | 77.6% | 74.3% | 75.0% | |

stem (*Figure 2C,D*; Watson–Williams test, (2C) *P*: $4.4 \times 10^{-16}$, n = 29 (left), 30 (right), (2D) *P*: 0.29, n = 26 (left), 10 (right); Rayleigh test, *P*: (2C) left, $3.1 \times 10^{-5}$, right, $8.2 \times 10^{-18}$, (2D) left, $2.0 \times 10^{-14}$, right, $5.1 \times 10^{-7}$). Considering that the barrier in the maze stem would have been an indication to the rats of the ongoing DA subtask, this suggests that the spatial memory demand in preparation for NA/DA subtasks decreased within the maze stem. In addition, prior to making erroneous choices, the rats did not make a decision at the behavioral level within the maze stem.

## Running path in the replay

I examined path representation in the temporally compressed replay of place cell activity sequences during the brief periods of immobility in the awake behavioral state. Some reactivations of place cell activity sequences during SWRs were related to the rats' upcoming path on the task. For instance, consider the spike raster plots of nine replays that occurred in the start zone, central stem, or junction (*Figure 3*). To arrange the sequences in terms of the cells' place fields along the journey, I ordered the cells by the latency of their peak firing rates during the corresponding trial type. The cell number decreases from the start zone to the goal. Sequential reactivation of the place cell activity during the replays shown in the raster plots began where the rat paused (blue circle in the trial-type illustration) and moved forward toward a goal.

To test whether the replays consistently related to upcoming behavior, I used the Bayesian decoder at a 10-fold compressed scale (*Davidson et al., 2009*; *Pfeiffer and Foster, 2013*) in the replay analyses. To avoid confusing replay and sequential activation during movement-related phase precession, population bursts of spiking activity during periods of awake immobility, when rats were moving less than 2 cm/s, were identified as candidate replays (*Figure 3—figure supplement 1A*). Of a total of 2820 candidates, 552 were identified as statistically significant replays with a continuous path, whose length and duration satisfied several criteria (*Davidson et al., 2009*; *Pfeiffer and Foster, 2013*) (see 'Materials and methods'; *Figure 3—figure supplement 1*; *Table 2*). LFPs during the path replays were largely coincident with the SWRs (99.8%; *Table 2*), and their theta power (4–12 Hz) was significantly smaller than during running (*Figure 3—figure supplement 2*). This confirms that the path replays identified here were similar to those reported previously (*Foster and Wilson, 2006*; *Diba and Buzsáki, 2007*; *Davidson et al., 2009*; *Pfeiffer and Foster, 2013*) and did not include low-speed replay events associated with theta oscillation (*Johnson and Redish, 2007*; *Jezek et al., 2011*).

To examine the spatial specificity of path representation in the replays, I analyzed the posterior probability of locations decoded from path replays by the Bayesian decoder. The middle portions of *Figure 3* show nine representative decoded paths from the spiking activity shown in the raster plots during candidate replays. As the spike raster plots suggest, the decoded paths also began where the rats paused and moved forward toward a goal in the corresponding trial type. In addition, despite the

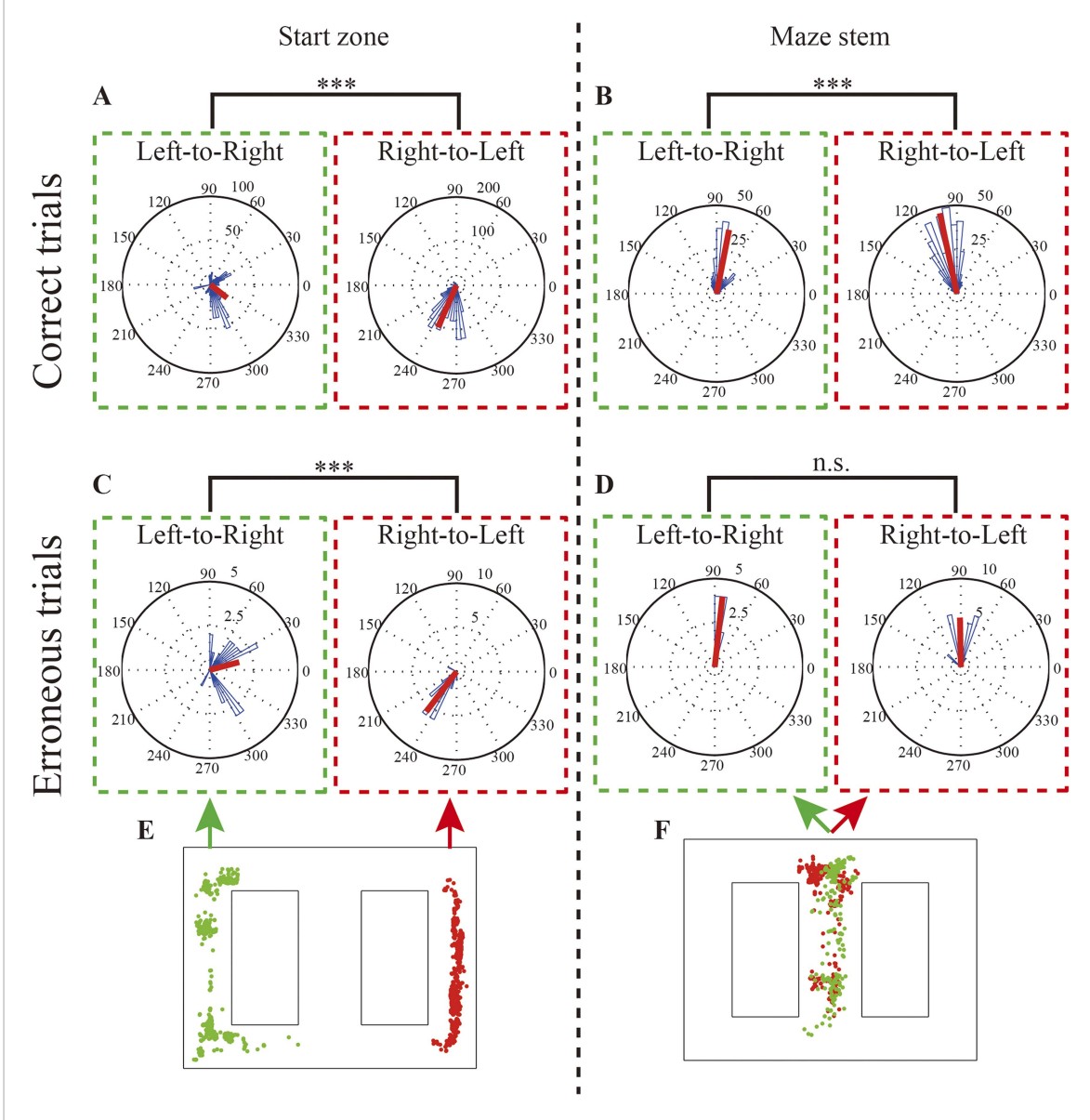

**Figure 2**. Rats' head direction during replays and replay location. (**A**, **B**) Rose diagrams of rats' head directions in replays occurring during left-to-right and right-to-left journeys in the start zone (**A**) and within the maze stem (**B**) in correct trials. (**C**, **D**) as for (**A**, **B**) but for erroneous trials. Occurrence locations are organized in two columns. In each column, green and red dotted lines enclose left-to-right and right-to-left journeys, respectively. (**E**, **F**) Physical locations where the replays occurred (green dots: left-to-right journeys; red dots: right-to-left journeys). Note that circular medians (red bars) were oriented toward a memory-guided goal (i.e., a goal opposite to the current location), except in replays occurring in the maze stem during erroneous trials. The head directions during delay periods in the DA subtask within the maze stem were excluded. ***: p < 0.0001, n.s: p > 0.05.

fact that the actual goal in the $VD_{RR}$ trial was on the right side, the decoded path ended on the left side (green arrow), suggesting that the replay represents an upcoming path to a memory-guided goal (i.e., opposite to the previous goal). Across all replays, the posterior probability of decoded locations was spatially concentrated around the vicinity of the goal (*Figure 4A*). To statistically compare the strength of path representation in the replays between selected regions, path representation strength was defined as the sum total of the posterior probability. Path representation in the vicinity of both goals was significantly greater than in the remainder of the maze area (*Figure 4B*; Wilcoxon rank-sum test, p = $3.3 \times 10^{-106}$, n = 552). As awake replay often begins at the animal's current location (*Foster and Wilson, 2006*; *Diba and Buzsáki, 2007*) and is enhanced by rewarding outcomes

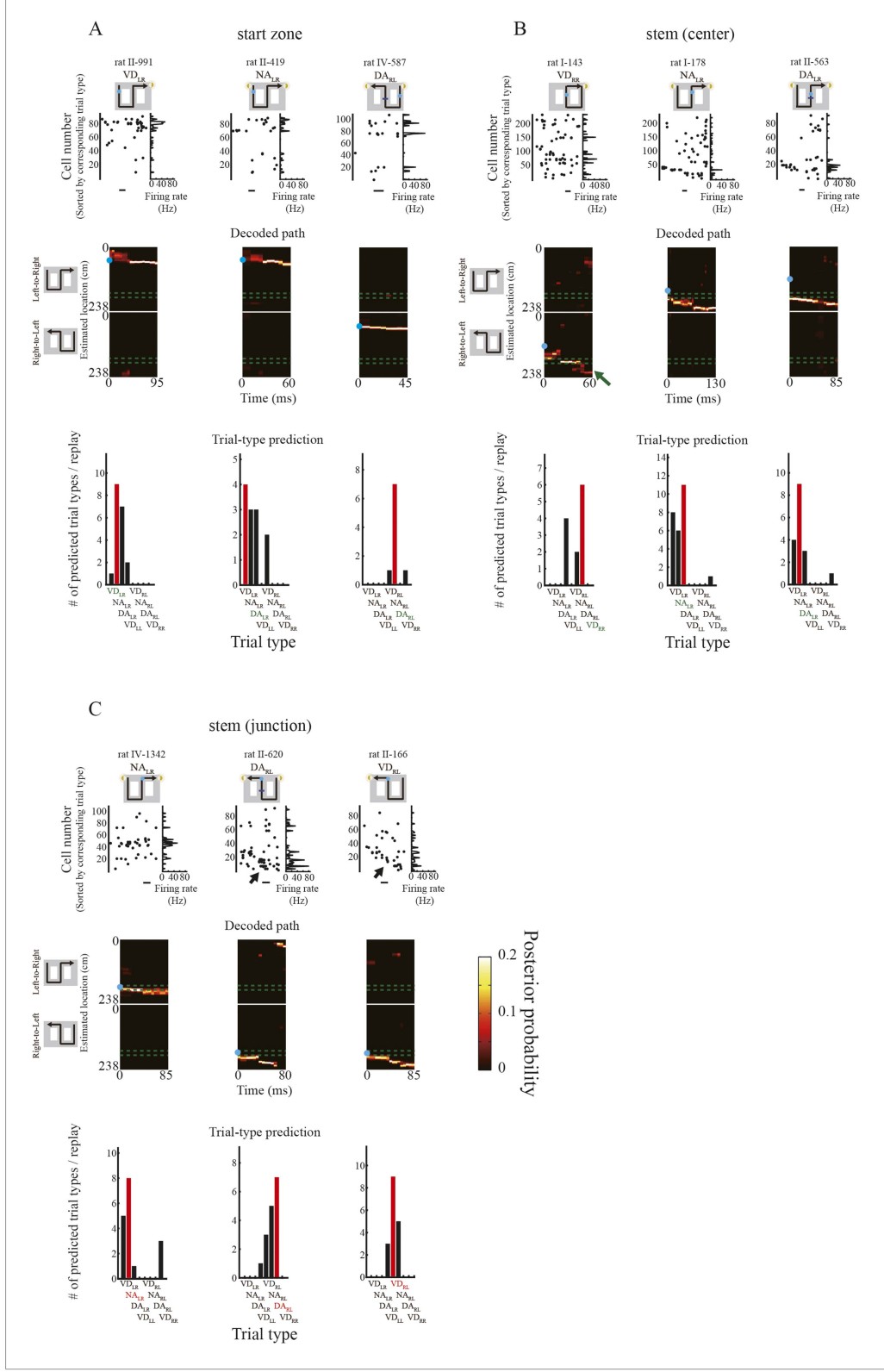

**Figure 3**. Decoding path and trial-type prediction during periods of awake immobility. Graphs are arranged in three columns according to the location at which the replay occurred (**A**: start zone; **B**: central maze stem; **C**: junction). Within each column is one subcolumn for each trial type. Each subcolumn consists of a raster plot of spiking activity of the place cells (top, left), the corresponding firing rates (top, right), the posterior probability of decoded paths

*Figure 3. Continued*

(middle), and predicted trial types (bottom) for representative candidate replays during periods of immobility. The scale bars indicate 10ms. Values are indicated by color bars (middle, right). In the decoded paths (middle), the upper junctions in the maze are enclosed by two green dotted lines, and the rat's physical location when the replay occurred is indicated by a solid blue circle. In the trial-type prediction (bottom), red bars indicate the most frequently predicted trial type. Red and green labels indicate whether the most frequently predicted and actual trial types matched or not, respectively. The replays depicted an upcoming path to a memory-guided goal irrespective of where the replay occurred. In fact, the path representation ended at a memory-guided, opposite goal to the visually guided goal in the VD$_{RR}$ (green arrow). During replays that occurred at the junction in DA$_{RL}$ and VD$_{RL}$ trials (rightmost, black arrows), the spike raster plots (top, left) and the decoded paths (middle) showed similar patterns. However, the firing rates (top, right) for DA$_{RL}$ were greater than those for VD$_{RL}$. Trial types predicted from the replays occurring at the junction (**C**, bottom) using the prediction method based on firing rates were accurate on the whole.

The following figure supplements are available for figure 3:

**Figure supplement 1**. Procedure for Bayesian decoding of locations and trial-type prediction.

**Figure supplement 2**. LFP theta power decrease in the replay.

**Figure supplement 3**. Estimated compression rate of firing during replays.

---

(**Singer and Frank, 2009**), the replay may be initiated at the goal. Although reward expectation increases occupancy time and place-field representation around the goals, such initiation bias was not correlated with either occupancy time (**Figure 4—figure supplement 1A,C**) or spatial distribution of place-field representation (**Figure 4—figure supplement 1B,D**). This suggests that the path replays were not a simple consequence of either reward expectation or place-specific firings in the location where they occurred (**Gupta et al., 2010**; **Carr et al., 2011**). Rather, they may represent paths on a memory-guided journey towards an upcoming goal (**Pfeiffer and Foster, 2013**), or on the immediately preceding journey (**O'Neill et al., 2006**; **Karlsson and Frank, 2009**).

## Replay represents a path to a memory-guided goal

To test this hypothesis, I examined the representation of future and past journeys in the replays. The path replays predominantly occurred in the start zone and in the maze stem (**Figure 2E,F**). Unlike the

**Table 2**. Replay statistics

| | Correct trials | | | | |
| | Candidate-start | | Candidate-stem | | |
|---|---|---|---|---|---|
| Number | 1893 | | 810 | | |
| Replay type | Path-start | Episodic-start | Path-stem | Episodic-center | Episodic-junction |
| Number | 345 | 71 | 190 | 17 | 17 |
| Percent confirmed | 18.2% | 20.6% | 23.5% | 21.5% | 15.3% |
| Percent SWR coincident | 100% | 100% | 99.5% | 98.7% | 100% |
| | Erroneous trials | | | | |
| | Candidate-start | | Candidate-stem | | |
| Number | 59 | | 58 | | |
| Replay type | Path-start | Episodic-start | Path-stem | Episodic-center | Episodic-junction |
| Number | 9 | 1 | 8 | 0 | 2 |
| Percent confirmed | 15.3% | 12.5% | 13.8% | 0% | 25.0% |
| Percent SWR coincident | 100% | 100% | 100% | – | 100% |

SWR, sharp wave/ripple.

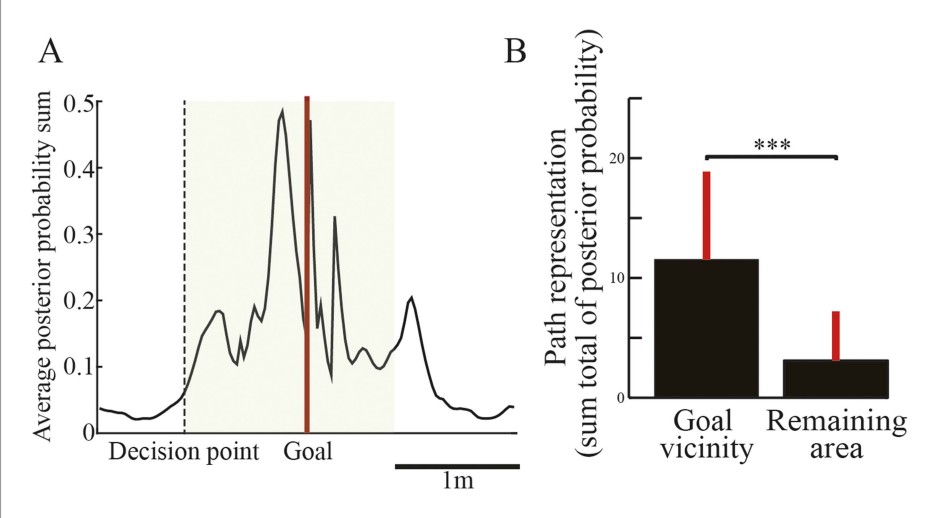

**Figure 4**. Spatial tendency of candidate replays. (**A**) Sum of posterior probabilities of decoded locations for the candidate replays throughout the entire session averaged across all rats. The dotted line indicates the location of the decision point, and the red line indicates the location of the goal. The green shaded area shows the defined goal vicinity. (**B**) Path representation, that is, the sum total of the posterior probabilities of decoded locations, within the goal vicinity and in the remaining area. Red lines indicate SD. ***: p < 0.0001.

The following figure supplement is available for figure 4:

**Figure supplement 1**. The relationship between occupancy time / place-field distribution and path replay representation.

start zones, the maze stem was common to all journeys. There were therefore two types of path replays: path-start and path-stem. I centered the posterior probability of decoded locations on the start position and rotated it according to the direction of the memory-guided goal. As expected from the initiation bias, the rats' current location was strongly represented in the posterior probability of decoded locations in path-start replays (*Figure 5A*). To statistically test the difference between future and past journeys, I defined future and past regions as those from the start position to memory-guided and previous goals, respectively. It is worth noting that the memory-guided goal was always opposite to the previous goal even in visually guided trials ($VD_{LL}$, $VD_{RR}$), because the task design did not allow the rats to identify the subtask until they reached the junction (see 'Materials and methods'). Path representation (i.e., the sum total of the posterior probabilities of decoded locations) of future regions was significantly greater than that of past regions in the correct spatial alternation trials (*Figure 5B*; Wilcoxon signed-rank test, p = 0.0037, n = 345). Similarly, future regions were strongly and statistically significantly represented in the correct visually guided trials (*Figure 5C,D*; p = 0.0078, n = 52). In erroneous trials, path representation of future regions was slightly greater than that of past regions (*Figure 5E*), but the difference was not significant (*Figure 5F*; p = 0.73, n = 9). However, due to the physically different start zones of different journeys, these characteristics may be affected by the place-specificity of the locations in which the replays occurred.

To compare the representation of past and future journeys in the path-stem replays, I centered the posterior probability of decoded locations on the rat's physical location at the time of occurrence, rotated it according to the direction of the memory-guided goal, and scaled it according to the distance to the memory-guided goal. Path representation was similar to that in the replays that occurred in the start zones (*Figure 5G,I*; *Figure 5H*: p = $9.8 \times 10^{-8}$, n = 190; *Figure 5J*: p = 0.034, n = 38). Surprisingly, however, path representation in path-stem replays on journeys toward an erroneous goal (*Figure 5K*) was significantly different from that on journeys toward a correct goal (*Figure 5L*; p = 0.0078, n = 8), despite the fact that the rats sampled both goal sides evenly. Similar to the findings of a previous study (*Pfeiffer and Foster, 2013*), this suggests that in the maze stem, where spatial memory retrieval is a prerequisite for predicting a future goal, path representation in the replay is directly linked to the animal's future path.

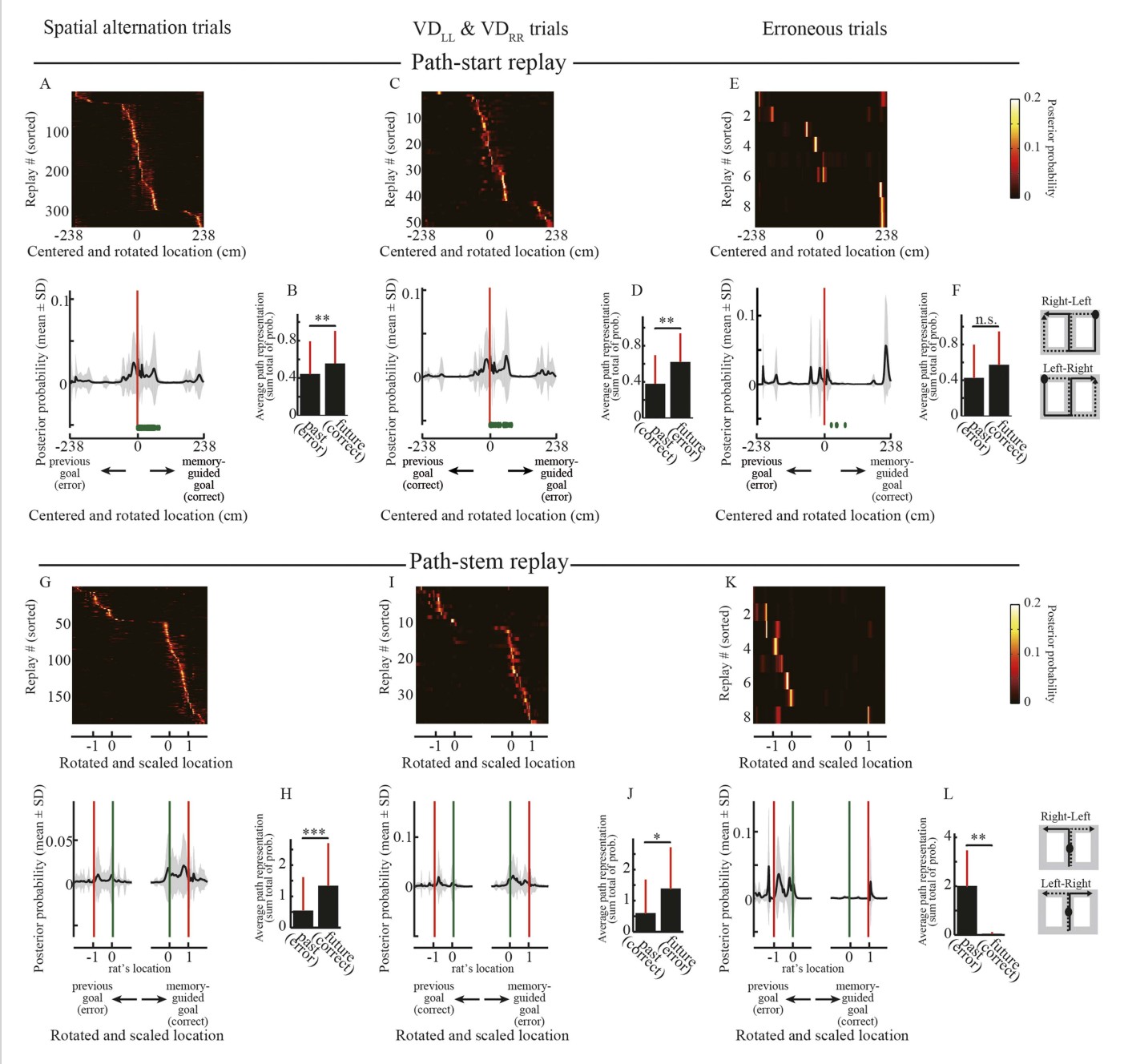

**Figure 5**. Decoded paths in path replays. (**A**, **C**, **E**) Upper panels show the posterior probability of decoded locations for path replays that occurred in the vicinity of the start position in the spatial alternation trials (VD$_{LR}$, VD$_{RL}$, NA$_{LR}$, NA$_{RL}$, DA$_{LR}$, DA$_{RL}$) (**A**), the visually guided trials (VD$_{LL}$, VD$_{RR}$) (**C**), and erroneous trials (**E**), centered on the start position and rotated according to the direction of the memory-guided goal. The replays were ordered by the location showing the maximum posterior probability. Values are indicated by color bars (right). Lower panels show that the posterior probability of the decoded location averaged across all path replays. Distance was centered on the start position and rotated according to the direction of the memory-guided goal. The shaded area indicates SD, green dots indicate the rat's physical location, and red lines indicate the start position. The schematic diagrams of the maze on the right show the start position (black filled circles) and rotation direction (solid line: towards the memory-guided goal; dotted line: towards the previous goal). Note that in the visually guided trials (VD$_{LL}$, VD$_{RR}$), the rats could get rewards at the previous goal but were always under spatial memory demand due to the task design, so the choice to go toward the memory-guided goal (i.e., the goal opposite to the previous goal) was defined as an erroneous response. (**B**, **D**, **F**) Path representation (the sum total of posterior probabilities of decoded locations) averaged across all path replays within future and past regions, defined from the start position to memory-guided/previous goals, for the trials shown in **A**, **C**, and **E**. Red bars indicate SD. (**G–L**) as for (**A–F**), but for path replays occurring within the stem. Distance was centered on the rat's physical location at the time of occurrence, rotated according to the direction of the memory-guided goal, and scaled according to the distance from the rat's physical location to a

*Figure 5. continued on next page*

*Figure 5. Continued*
memory-guided goal. Green lines indicate the rat's physical location, and red lines indicate the memory-guided/previous goals. Future and past regions were defined based on the rat's physical location and the memory-guided/previous goals, respectively. *: p < 0.05, **: p < 0.01, ***: p < 0.001, n.s.: p > 0.05.
The following figure supplement is available for figure 5:

**Figure supplement 1**. Decoded paths with the highest a posteriori probability in path replays.

The exact path encoded in the replay may be decoded from the locations showing the highest a posteriori probability. I therefore reanalyzed the path replays using point estimates of locations based on a posteriori probability. Similar to the posterior probability analyses, the decoded locations in the replays tended to represent an upcoming journey to a memory-guided goal (*Figure 5—figure supplement 1*). To corroborate the results from the Bayesian decoder, I furthermore applied a simple spatial reconstruction algorithm, based on place fields during running, to the path replays. To estimate path representation in the replay, the place maps of the maze in neurons participating in the replay were simply summed up in response to the number of spikes (see 'Materials and methods'). Similarly to the results obtained using the Bayesian decoder, the path representation estimated using the simple spatial reconstruction algorithm also largely predicted an upcoming path to a memory-guided goal (*Figure 6*). A previous study (*Pfeiffer and Foster, 2013*) reported that the awake replays represent an upcoming path to a memory-guided goal, but did not examine erroneous behaviors. My findings strongly support this view, with the additional finding that the upcoming path encoded in the awake replay can be linked to the animal's future actions.

## Replay is enhanced by spatial working memory demand

Although hippocampal place cell activity per se represents non-spatial information as well as spatial information (*Wood et al., 1999*), the question of whether replays include non-spatial information remains to be addressed. I next examined whether place cell activity during replays can convey information not only on the path but also on the subtask occurring along it, similarly to place cell activity occurring during running.

I observed that the firing rate of place cells participating in replays during different subtasks was dramatically different between subtasks even if the replay depicted a similar decoded path. For instance, consider the firing rate histograms for participating neurons in replays occurring in the start zone, central stem, or junction (*Figure 3*). As expected from the timing of temporally compressed firings across place cells, the maximum firing rates of place cells during replays (∼80 Hz) were about 10 times faster than typical maximum firing rates during running (∼8 Hz). To estimate the exact temporal compression rate of place cell firing during replays, I divided the average interspike interval during candidate replays by that during the entire rest of the session. The median of the estimated compression rate was 9.0 (*Figure 3—figure supplement 3*), suggesting that similarly to the timing of firing, place cell firing rates were also temporally compressed during replays. During replays occurring at the junction, the raster plots of spiking activity in $DA_{RL}$ and $VD_{RL}$ trials recorded from rat II showed a similar activity sequence (*Figure 3C*, rightmost, arrows). The replays also depicted a similar decoded path. However, the firing rates during $DA_{RL}$ trials were substantially greater than during $VD_{RL}$ trials, suggesting that the firing rates of neurons participating in the replay may contain information on the differences between subtasks.

To test whether subtask information is encoded in the firing rates in the replay, I predicted trial types based on the firing rates of a group of cells recorded during path replays, using the prediction method in conjunction with the decoded locations (see 'Materials and methods'; *Figure 3—figure supplement 1*). It was necessary to ensure that the replays used predicted a trial type with a probability greater than that dictated by chance. Accordingly, of the total 552 path replays, 108 were classified as episodic replays that satisfied criteria concerning the most frequently predicted trial type and its predicted probability sum (see 'Materials and methods'; *Table 2*). As expected, only the replays that occurred in the junction (*Figure 3C*) appeared to predict the actual trial type correctly.

To examine this further, I investigated preference for subtask (i.e., VD, NA, DA subtasks). I examined whether there was any preference in the replays for particular rewarded subtasks. As

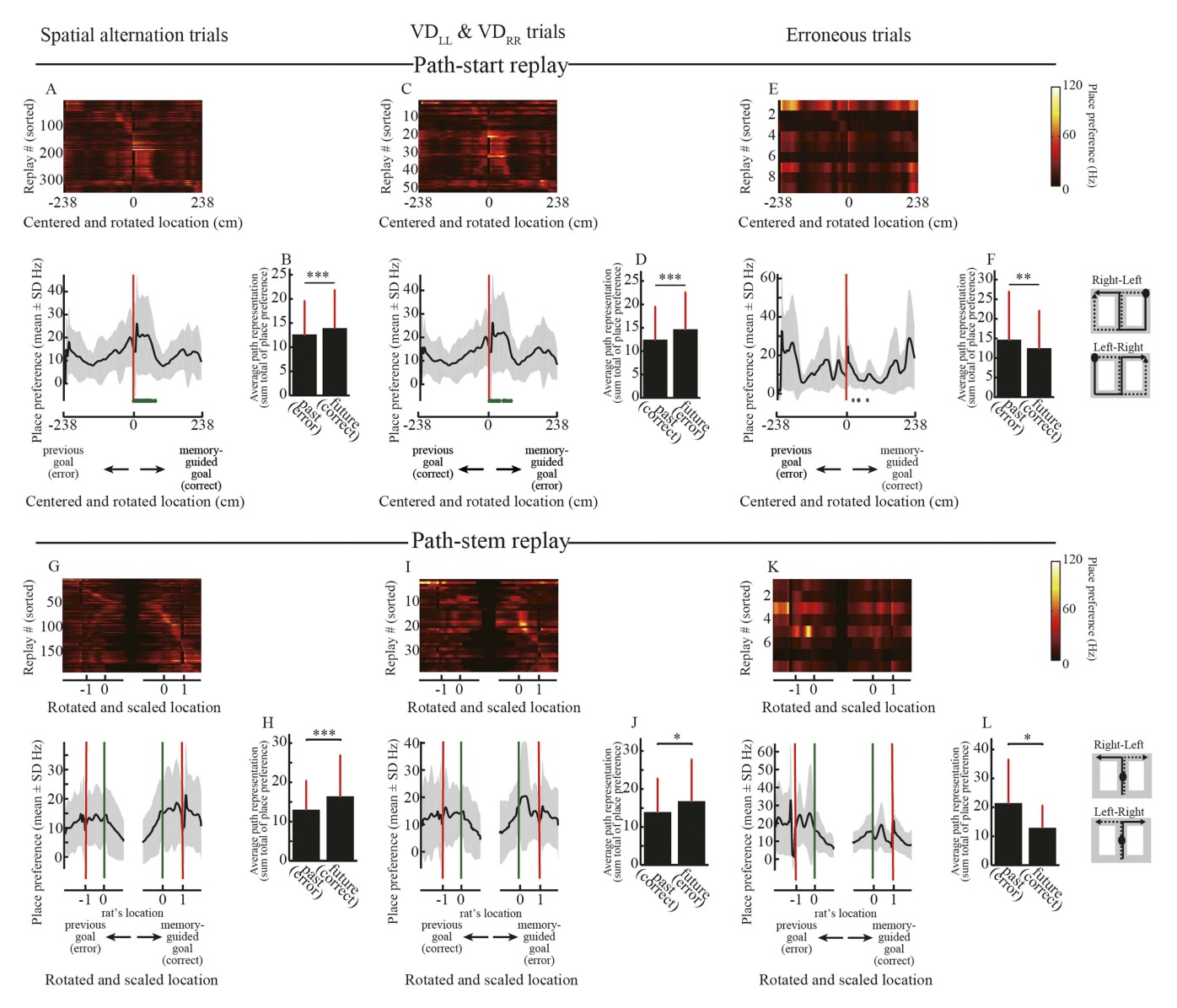

Figure 6. Estimated place preference in the path replays. (A, C, E) Upper panels show the place preference estimated using the simple spatial reconstruction algorithm for path replays that occurred in the vicinity of the start position in the spatial alternation trials (VD$_{LR}$, VD$_{RL}$, NA$_{LR}$, NA$_{RL}$, DA$_{LR}$, DA$_{RL}$) (A), the visually guided trials (VD$_{LL}$, VD$_{RR}$) (C), and erroneous trials (E), centered on the start position and rotated according to the direction of the memory-guided goal. The replays were ordered by the location showing the maximum place preference. Values are indicated by color bars (right). Lower panels show that the place preference averaged across all path replays. Distance was centered on the start position and rotated according to the direction of the memory-guided goal. The shaded area indicates SD, green dots indicate the rat's physical location, and red lines indicate the start position. The schematic diagrams of the maze on the right show the start position (black filled circles) and rotation direction (solid line: towards the memory-guided goal; dotted line: towards the previous goal). Note that in the visually guided trials (VD$_{LL}$, VD$_{RR}$), the rats could get rewards at the previous goal but were always under spatial memory demand due to the task design, so the choice to go toward the memory-guided goal (i.e. the goal opposite to the previous goal) was defined as an erroneous response. (B, D, F) Path representation (the sum total of place preferences) averaged across all path replays within future and past regions, defined from the start position to memory-guided/previous goals, for the trials shown in A, C, and E. Red bars indicate SD. (G–L) as for (A–F), but for path replays occurring within the stem. Distance was centered on the rat's physical location at the time of occurrence, rotated according to the direction of the memory-guided goal, and scaled according to the distance from the rat's physical location to a memory-guided goal. Green lines indicate the rat's physical location, and red lines indicate the memory-guided/previous goals. Future and past regions were defined based on the rat's physical location and the memory-guided/previous goals, respectively. Wilcoxon signed-rank test, *: P<0.05, **: P<0.01, ***: P<0.001.

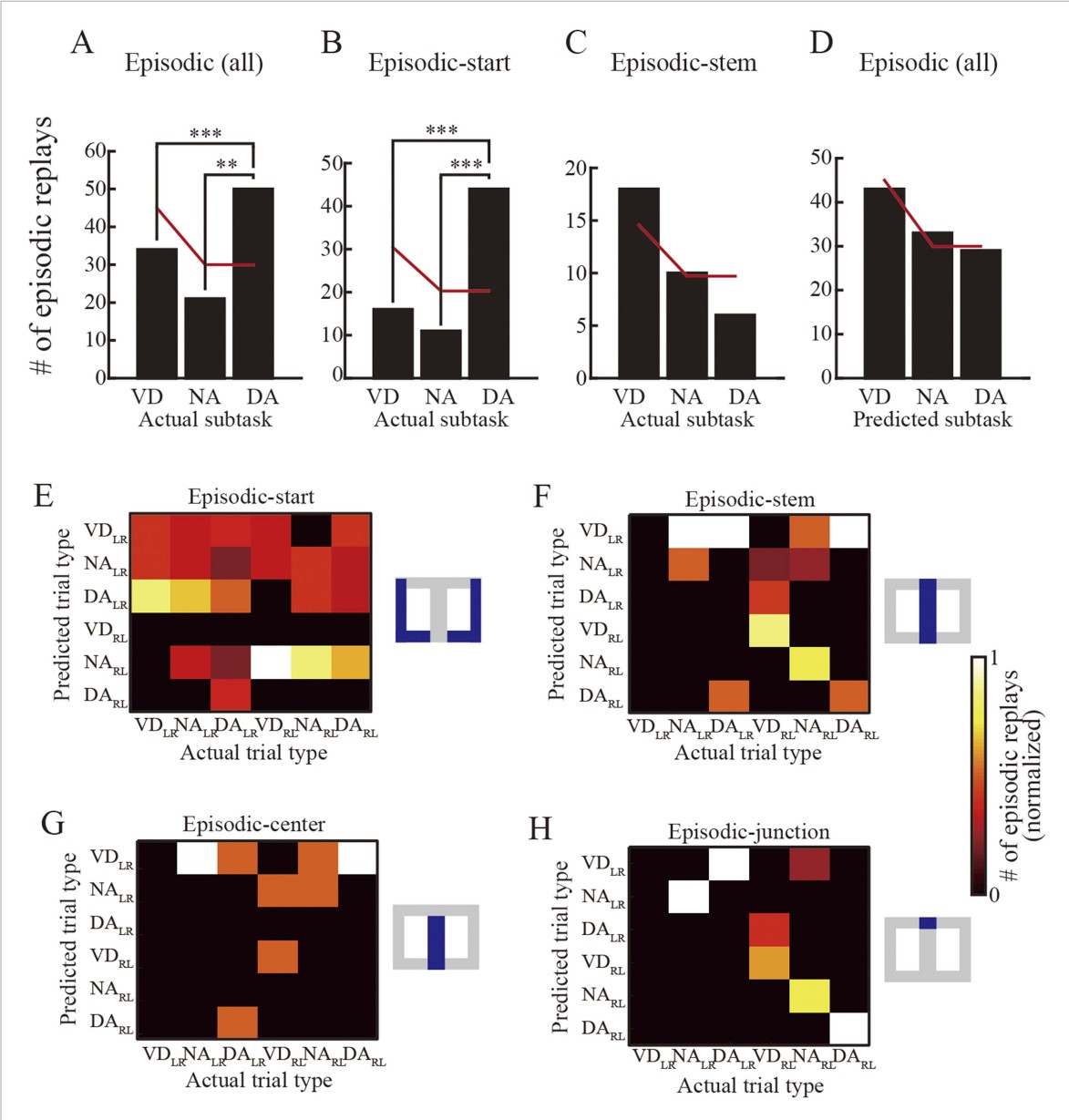

**Figure 7**. Subtask preference and relationship between actual and predicted trial types. (**A**) The number of episodic replays across all rats as a function of actual subtasks to be rewarded. Red lines indicate the estimated numbers calculated by multiplying the total number of replays by the ratio of repeatedly experienced subtasks throughout the entire session (60: 40: 40; null hypothesis: the replays represent all subtasks evenly). (**B**, **C**) The number of episodic-start (**B**) and episodic-stem (**C**) replays across all rats as a function of actual subtasks. Red lines as in (**A**). (**D**) The number of episodic replays across all rats as a function of subtasks predicted from the replay. **: $p < 0.001$, ***: $p < 0.0001$. (**E–H**) The number of episodic replays across all rats normalized across predicted trial types of matches between actual and predicted trial types (for spatial alternation trials only) in episodic-start (**E**), episodic-stem (**F**), episodic-center (**G**), and episodic-junction (**H**) replays. Blue in the schematic diagrams to the right indicates the regions in which the replays occurred in the maze. Note that whereas $VD_{RL}$ and $NA_{LR}$ trials were frequently predicted from the episodic replays in the central stem when the actual trial type was $VD_{RL}$ (**G**), such partial matching may be accidental. In contrast, in the junction (**H**), the trial types showing the largest predicted value matched the actual trial types well for $NA_{LR}$, $VD_{RL}$, $NA_{RL}$, and $DA_{RL}$ trials. The overall matching score across all trial types is significantly different from random matching (z test, $p = 3.7 \times 10^{-4}$), suggesting that episodic replays can accurately represent actual trial types, but only in the junction.

The following figure supplements are available for figure 7:

**Figure supplement 1**. Subtask preference in replays occurring in the start zone, and time spent pausing.

*Figure 7. continued on next page*

*Figure 7. Continued*

**Figure supplement 2**. Relationship between actual and predicted trial types using the simple spatial reconstruction algorithm.

**Figure supplement 3**. Trial-type prediction on the time-window basis.

expected, based on a previous study suggesting that awake SWRs support spatial working memory and not reference memory (*Jadhav et al., 2012*), the occurrence rate of overall episodic replays across all actual subtasks to be rewarded was significantly biased toward the DA subtask, which required spatial working memory (*Figure 7A*; Chi-square goodness-of-fit test, post hoc multiple binomial tests corrected by the Benjamini and Hochberg procedure, VD vs DA: $p = 8.6 \times 10^{-5}$, NA vs DA: $p = 9.9 \times 10^{-4}$). The null hypothesis tested was that the occurrence rates of predicted subtasks were equal to the rates expected based on the ratio of subtasks experienced throughout the entire session (60 laps VD: 40 laps NA: 40 laps DA).

Does this bias depend on where the replay occurred? To address this question, episodic replays were divided into two types according to the occurrence location of the replay: episodic-start and episodic-stem replays. This was done because the head direction analyses suggested that spatial memory demand at the behavioral level was lower within the maze stem. The partitioning showed that the subtask bias occurred in episodic-start replays (*Figure 7B*; VD vs DA: $p = 3.7 \times 10^{-9}$; NA vs DA: $p = 5.1 \times 10^{-7}$), but not in episodic-stem replays (*Figure 7C*). Similarly, in the path-start replays, all candidates and SWR events showed such a bias (*Figure 7—figure supplement 1A–C*). This bias may be accounted for by occupancy time while the rats paused. However, since the rats specifically spent more time pausing in the DA subtask in both the start zones and the maze stem (*Figure 7—figure supplement 1D,E*), the bias cannot be explained by the time spent pausing. Combined with behavioral signs of decreased spatial memory demand within the maze stem, these results suggest that the occurrence rate of replays is enhanced by upcoming spatial working memory demand. In contrast, the subtasks predicted from the episodic replays did not show such a bias, and instead the prediction rates reflected the ratio of subtasks experienced throughout the entire session (*Figure 7D*). This suggests that the enhancement is not specifically due to the reactivated DA subtask, but instead due to the reactivation of all previously experienced subtasks.

## Replay can predict the actual trial type to be rewarded

Examination of trial type, composed of journey type and subtask, should provide further understanding of subtask preference. Even if the replay can represent an upcoming path, disambiguation of the subtasks is still required for identifying the actual trial type. I therefore investigated the relationship between the actual trial type to be rewarded and trial-type representation in the episodic replays. Since the path representation suggested that the replay encodes an upcoming path to a memory-guided goal, I only examined the memory-guided trial types (i.e., $VD_{LR}$, $VD_{RL}$, $NA_{LR}$, $NA_{RL}$, $DA_{LR}$, and $DA_{RL}$). Trial-type representation in the episodic replays did not match the actual trial type in the start zones (*Figure 7E*; Choen's $\kappa = 0.017$; z test, $p = 0.83$, $n = 71$), but did match it in the stem (*Figure 7F*; Choen's $\kappa = 0.31$; $p = 8.4 \times 10^{-3}$, $n = 34$). This discrepancy may be the result of the extent of spatial working memory demand at the location where the replay occurs.

To further examine trial-type preference, I divided the episodic-stem replays into two types: episodic-center and episodic-junction replays. Trial-type representation matched actual trial type moderately well in replays at the junction (i.e., the decision point) where the rats could evidently identify the actual trial type based on the visual cue (*Figure 7H*; Choen's $\kappa = 0.58$; $p = 3.7 \times 10^{-4}$, $n = 17$), but not at the center of the stem (*Figure 7G*; Choen's $\kappa = 0.022$; $p = 0.93$, $n = 17$). To corroborate these results, the Bayesian decoder in the trial-type prediction method was replaced by the simple spatial reconstruction algorithm. Similarly, trial types predicted using this method matched actual trial types moderately well only in replays at the junction (*Figure 7—figure supplement 2*). To further check whether the predicted and actual trial types were coincidently matched, the probability of trial-type prediction across all replays occurring at the junction was analyzed on a time-window basis.

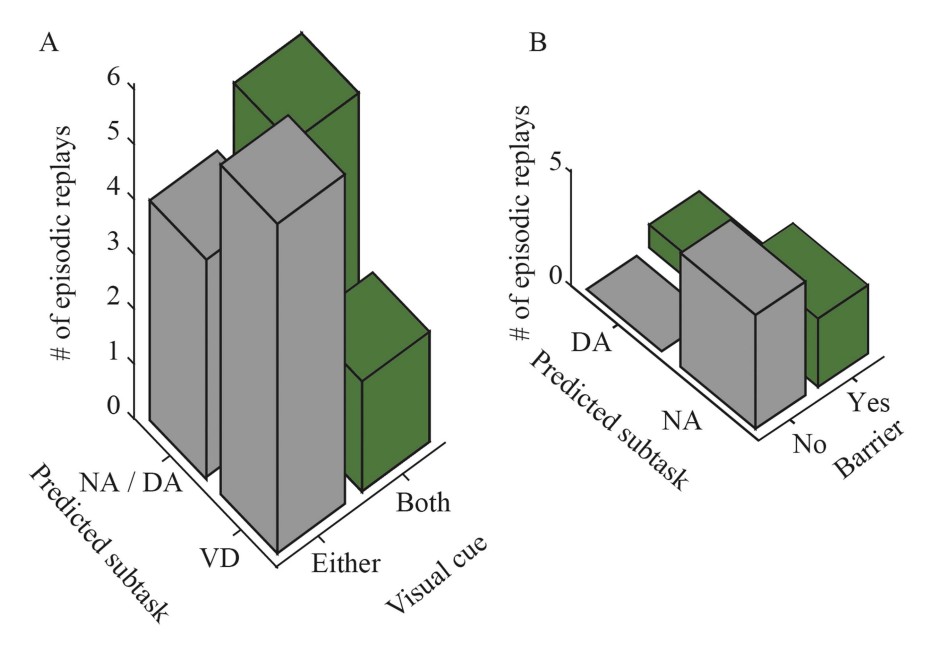

**Figure 8**. The relationship between predicted subtask and sensory cue. (**A**) The number of episodic-junction replays with matches between the predicted subtask and the configuration of visual cues, across all rats. (**B**) The number of episodic-center replays with matches between the predicted memory-guided subtask and the presence of a barrier at the center of the maze, across all rats.

There was a significant effect of the probability of trial-type prediction for five trial types ($NA_{LR}$, $DA_{LR}$, $VD_{RL}$, $NA_{RL}$, and $DA_{RL}$) in episodic replays (ANOVA, $p < 0.01$; *Figure 7—figure supplement 3*). The predicted trial types showing the highest probability matched actual trial types for $NA_{LR}$, $VD_{RL}$, $NA_{RL}$, and $DA_{RL}$ trials. Post hoc comparisons using the Tukey honest significant difference (HSD) test indicated that the predicted trial types with the highest probability were significantly different from most of the others, suggesting that the matches were not coincidental.

At the junction, only a single visual cue was present in the visually guided trials. To test whether the trial type encoded in these replays was therefore just a representation of the available sensory cues, I examined whether the difference in the number of visual cues showing (one or both) could be used to distinguish VD from the other subtasks encoded in replays at the junction. The number of visual cues showing did not account for VD representation, even at the junction (*Figure 8A*; $\chi^2$ test, $p = 0.20$). The barrier that appeared at the center of the stem only in DA trials may also represent a sensory cue, which would distinguish NA from DA trials. However, the subtask predicted in the center of the stem did not match actual NA and DA trials (*Figure 8B*; $\chi^2$ test, $p = 0.26$). This suggests that trial-type representation in the replays is more than just a representation of external cues, and that it is internally generated. I therefore conclude that the internally generated replay represents the actual trial type—not only the upcoming path but also the actual subtask to be rewarded—when the trial type can be identified, but otherwise it evenly represents all trial types thus far experienced on the reactivated path.

## Discussion

My investigations into whether the subtask and path experienced during task performance are represented in replays of place cell activity sequences associated with SWRs during brief immobility periods, produced the following findings. Firstly, the replay can represent non-spatial information on subtask as well as spatial information concerning the path. Secondly, whereas the path in the replay is encoded in temporally compressed firing timings across place cells, the accompanying subtask is encoded in their temporally compressed firing rates. Thirdly, the replay is enhanced by spatial working

memory demand. Finally, the replay can represent the actual trial type accurately when the trial type can be identified, but otherwise it evenly represents all trial types previously experienced along the reactivated path.

Place cell activity can encode not only spatial information about past, current, and future places (*Frank et al., 2000*; *Wood et al., 2000*; *Ferbinteanu and Shapiro, 2003*) but also non-spatial information including task demand (*Markus et al., 1995*; *Anderson and Jeffery, 2003*; *Smith and Mizumori, 2006*), odors and their match/non-match status (*Otto and Eichenbaum, 1992*; *Sakurai, 1996*; *Wood et al., 1999*), and elapsed time (*Pastalkova et al., 2008*; *MacDonald et al., 2011*). The sequential activation of place cells reflects distinct episodic-like memories (*Pastalkova et al., 2008*); in fact, reactivation of the firing rates of single neurons in the human hippocampus represents specific episodes during free recall (*Gelbard-Sagiv et al., 2008*). However, such episodic-like memory traces are present across time-scales similar to those during the actual experience, and are thus distinct from the rapid, temporally compressed replays reported here. Although numerous previous reports have concluded that place cell activity sequences are reactivated in a temporally compressed manner in the awake state (*Foster and Wilson, 2006*; *O'Neill et al., 2006*; *Diba and Buzsáki, 2007*; *Singer and Frank, 2009*; *Gupta et al., 2010*) as well as during slow-wave sleep (*Wilson and McNaughton, 1994*), they have shown only that replays encode familiar paths. Until now, the possibility that replays simultaneously represent both spatial and non-spatial information has been purely conjectural. It is well known that spatial information is encoded in temporally compressed firing timings across place cells in replays. I have now shown that replays also encode non-spatial information in temporally compressed firing rates, revealing that global and rate remapping mechanisms during running (*Leutgeb et al., 2005*; *Takahashi, 2013*) are preserved in the temporally compressed replays that occur during brief periods of immobility.

Since awake replay is more common in a novel environment (*Cheng and Frank, 2008*), a few previous studies (*O'Neill et al., 2006*) have speculated that the content of replays reflects the total of previous experience. Contradicting with this hypothesis, growing evidence suggests that replays are not a simple function of experience (*Gupta et al., 2010*), but rather that reward outcomes enhance the reactivation of experiences (*Singer and Frank, 2009*) and that replays can depict future paths to remembered goals (*Pfeiffer and Foster, 2013*). In addition, awake replays often begin at the animal's current location (*Diba and Buzsáki, 2007*). The relationship between replay content and experience therefore does not seem to be straightforward. Some previous reports suggest that awake replay is similar to vicarious trial-and-error (VTE) events (*Hu and Amsel, 1995*; *Johnson and Redish, 2007*), because it can represent possible future paths even in completely unfamiliar places (*Gupta et al., 2010*). Although VTE events are reported to occur outside of SWRs, a subsequent study suggested that awake replay associated with SWRs is seen frequently at the choice point in a similar task, where animals must make a memory-guided decision between two journeys (*Karlsson and Frank, 2009*). Consistent with activity observed in VTE events, these awake replays extend out to the left or right arms of the track, as would be expected if the animal were playing out possible upcoming choices. Frank et al. therefore proposed a unified interpretation in which replays provide information on possible upcoming paths to downstream brain structures such as the prefrontal cortex and nucleus accumbens, which can assess the value of different paths and make a decision about future actions (*Carr et al., 2011*).

In this study, I trained rats to perform a similar task, but path representation in these replays was directly linked to the animal's future path. This apparent disparity may reflect the proficiency of the task in previous studies. In contrast to previous VTE studies (*Johnson and Redish, 2007*; *Karlsson and Frank, 2009*; *Gupta et al., 2010*), well-trained rats in the present study could accurately predict a goal-directed path at the behavioral level even at the decision point. However, the task design did not allow prediction of the subtask until the animal had entered the decision point. This suggests that not only the path but also the subtask representation in the replay is similar to VTE events. By extending the unified interpretation to include non-spatial memory in the retrieved memory, the fact that replays in the present study reflected all previously experienced trial types (except at the decision point) suggests that replays prepare the brain for unforeseen changes, which would allow continuous switching between subtasks.

I also found that replays were enhanced by spatial working memory demand. Since the initiation of an awake replay is often related to external spatial input at the animal's location (*Karlsson and Frank, 2009*),

this enhancement of replays suggests that the internal demand of the hippocampal-dependent spatial working memory is another factor. Considering that this is similar to a cued memory retrieval process, my findings strongly support the hypothesis that awake replay plays a key role in episodic-like memory retrieval, which contributes to memory-based navigational planning and decision making (*Carr et al., 2011*).

The sequential activation of place cell activity associated with SWRs can also represent paths where the animal has never experienced (*Dragoi and Tonegawa, 2011*; *Dragoi and Tonegawa 2013a*; *2013b*; *Ólafsdóttir et al., 2015*). As a result, a fairly large proportion of events representing different and multiple experiences can spontaneously occur, even in naive animals. I speculate that the reason why paths and subtasks were encoded in temporally compressed firing timings and rates, respectively, in the replays may be to arrange such multiple preplayed episodes into a single replay event to easily recall and imagine an upcoming episode based on prior experience. In the present study, the rats were sequentially trained in the VD, NA, and DA subtasks. Together with the preplay mechanism, this training protocol might have helped to recruit a consistent path representation across the three subtasks. Until now, previous studies investigating the content of preplays have primarily focused on the geography and timeline of the animals' running. Non-spatial information in the preplay may provide further insight into memory encoding and retrieval.

This study has shown that non-spatial subtask information occurring along temporally ordered places, as experienced by the animal, could be predicted by the sequential reactivation of place cells occurring while the animal paused during engagement in the task. I therefore speculate that awake replay is linked to the typical abilities of episodic memory: mental time travel and foreseeing future situations. This provides novel insight into debates on animals' capacities to use the faculties provided by episodic memory (*Allen and Fortin, 2013*).

## Materials and methods

### Animals

Four male Wistar rats implanted with a custom-made microdrive were housed individually in cages where the light was maintained in a 12-hr light–dark cycle. The tests were performed during the light phase. The weight of all rats was kept at 80% of free-feeding body weight. All procedures were approved by the Doshisha University and Kyoto Sangyo University Institutional Animal Care and Use Committees.

### Protocol

The rats were initially trained to run along the left/right O-shaped track unidirectionally for reward signals (medial forebrain bundle (MFB) stimulation) in the start zone. After the rats were running smoothly along both left and right O-shaped tracks, they were trained in the figure-eight maze (overall: 100 × 140 cm, 20-cm height; path: 20-cm width; *Figure 1A*) to perform the VD subtask for a reward. In the VD subtask, one of the two visual cues (LED-lights at the right and left corners) was illuminated randomly; the visual cue at the decision point of the maze (*Figure 1A*, decision point) indicated which direction to turn to receive the reward. The rats were trained until they achieved at least 80% correct decisions for >20 laps. Next, they were trained to perform the NA subtask for >20 laps. In this case, both LEDs were illuminated, so the rats could not rely on the visual cue. Instead, they had to choose the direction opposite to the previous goal. Once they had achieved at least 80% correct decisions for >20 laps, they were trained to perform the DA subtask for 20 laps. This was almost identical to the NA subtask except that it included a delay period. This was enforced by a barrier appeared for 5 s, 20 cm ahead of the entrance to the maze stem (*Figure 1A*, red dotted line). The rats paused reliably, facing the forward direction, in front of the barrier, and achieved a 90% correct decision rate from the beginning of this task. The detailed training protocols are described in a previous study (*Takahashi, 2013*).

To prevent the rats from receiving any unintended distal room cues, the task was performed under dim light. They did not receive the visual cue until they reached the decision point because the height of the walls of the maze (20 cm) was much higher than the level of the rat's eyes (~5 cm) and they did

not stand up during the experiment. The sole signal for subtask switching was therefore LEDs that were visible only at the decision point.

## Surgery, electrode preparation, and recording

Under isoflurane anesthesia, stimulation electrodes were inserted into the MFB in the right lateral hypothalamus (AP 2.5, ML 1.0, DV 9.5). A custom-made microdrive with 10 independently movable dodecatrodes was then fixed to the skull above the left hippocampus (AP 3.8, ML 3.0, DV 0.5). A week after surgery, the electrodes were individually lowered into the pyramidal cell layer of the hippocampal CA1. The extracellular signals were unity-gain buffered, filtered (600 Hz–6 kHz), amplified (gain = 5000), and continuously sampled at 25 kHz. One channel of each electrode was duplicated for detecting LFPs and was filtered (0.1 Hz–6 kHz) and amplified (gain = 500). The LFP recordings were further band-pass filtered (150–250 Hz) using the Hilbert transform. Data recorded during the MFB stimulation were removed prior to analysis. Local peaks in the power of this filtered signal of magnitude >mean + 3SD were identified and extracted as periods. The boundaries for each period were defined as the point at which the amplitude crossed the mean. Periods during which running speed was <2 cm/s were identified as SWR events. For theta-wave detection, the z-score power of a band-pass filtered (4–12 Hz) signal was calculated. After spike sorting as previously described (*Takahashi et al., 2003*; *Takahashi and Sakurai, 2009*; *Takahashi, 2013*), putative principal cells were distinguished from putative fast-spiking cells based on spike width (0.4 ms) and average firing rate (5 Hz). Cells whose firing rate was <0.1 Hz were excluded. Only the putative principal cells were used in the analyses described below. Spikes during delay periods in the DA subtask, when hippocampal activity was not place-specific (*Pastalkova et al., 2008*), were excluded.

## Place map

The trajectory of the rats was linearized for each trial by projecting the actual trajectory onto a predefined idealized journey using nearest-neighbor Delaunay triangulation. Spatial bins had a resolution of approximately 2 cm. A one-dimensional map of the place field for each place cell was then constructed for each trial type in the standard manner (*Takahashi, 2013*) (smoothed with a Gaussian kernel of 3-cm width). The place map was constructed from spikes only when running speed was >5 cm/s so that spikes generated while the rats paused in front of the barrier wall in the DA subtask were excluded.

## Bayesian decoding

A memoryless Bayesian decoder (*Zhang et al., 1998*) was used to decode the rats' locations on the basis of place cell activity. Firstly, the probability of a rat's location given place cell firings within a time window was estimated as follows:

$$\mathrm{Prob}(Pos|spikes) = \left( \prod_{i=1}^{N} f_i(Pos)^{n_i} \right) \exp^{-\tau \sum_{i=1}^{N} f_i(Pos)}$$

where $f_i$ and $n_i$ represent the place map and the number of spikes of the $i$-th place cell within the time window, respectively, $N$ indicates the total number of place cells, and $\tau$ represents the duration of the time window.

The probability within each time window was normalized for every location as follows (*Pfeiffer and Foster, 2013*):

$$\mathrm{nProb}(Pos|spikes) = \mathrm{Prob}(Pos_k|spikes) \left/ \sum_{k=1}^{M} \mathrm{Prob}(Pos_k|spikes) \right.$$

where Prob($Pos_k$|spikes) represents the probability at the $k$-th location bin within the time window, and $M$ represents the total number of location bins.

Time windows for task performance and replays were set at 250 ms and 20 ms, respectively. Provided that nProb($Pos_k$|spikes) showed unimodality (Hartigan's dip test, $p < 0.05$), a point

estimation of the location on the journeys was made based on this value, using maximum likelihood estimation.

## Simple spatial reconstruction algorithm

A simple spatial reconstruction algorithm was used to virtually reconstruct the place preference of replays from place cell activity while the animal briefly paused. The place preference of replays, $Rp$, was estimated as follows:

$$Rp = \sum_{i=1}^{N} n_i f_i$$

where $f_i$ and $n_i$ represent the place map and the number of spikes of the $i$-th place cell firing in the replay, respectively, and $N$ indicates the total number of place cells. For trial-type prediction, the location encoded in the replay was estimated based on the location where $Rp$ was maximized.

## Path replay detection

Firstly, candidate replays were selected as follows (as described in *Figure 3—figure supplement 1A*). A candidate replay was defined as a period where the smoothed (Gaussian kernel; SD: 10 ms) population activity was greater than the mean and the peak was above the defined threshold (mean + 3SD). To minimize ambiguous detections, periods were decreased based on the following criteria. Time windows were reduced as much as possible such that at least 10% of all detected cells were detected within the time window, each cell included at least two spikes, and the duration was at least 30 ms. Secondly, for each period, the rat's location was decoded using the Bayesian decoder, for 20-ms time windows advanced in 5-ms increments. Periods were concatenated when the distance between neighboring decoded locations was less than 25 cm. Thirdly, concatenated periods whose sequence of decoded locations covered a total distance greater than that covered during four time windows were classified as candidate replays. Fourthly, two significance tests were conducted for every candidate, using a Monte Carlo method with two different random shuffling modes: cell identity and cell place field (5000 times each). Finally, candidates with p < 0.05 for both shuffle modes were defined as path replays.

## Trial-type prediction

Trial type was predicted for every time window, using mean firing rate as a point of reference (*Allen et al., 2012*). Firstly, a binary population vector, $C$, was constructed. This vector could exist in one of two states, high or low. The state was determined on the basis of 200-ms time windows throughout the entire session, based on whether the cell's firing rate in each time window was higher or lower than its mean firing rate for the whole session. Eight population reference vectors, $R$, were constructed to store the probability of the place cells being in the high or low state in each of the eight trial types, for every possible linearized location. The probability of obtaining the vector $C$ within each time window was computed as follows:

$$\mathrm{Prob}(C|t) = \prod_{i=1}^{N} \mathrm{Prob}_i(c|t)$$

where $\mathrm{Prob}_i(c|t)$ represents the probability, taken from vector $R$, that the $i$-th place cell is in state $c$ during the $t$-th trial type, at location $p$, decoded from place cell activity within the corresponding time window using the Bayesian decoder or the simple spatial reconstruction algorithm. For each time window, the trial type with the highest value for $\mathrm{Prob}(C|t)$ was recorded as predicted for that time window.

## Episodic replay detection

For each path replay, trial type was predicted for each time window. The representative trial type for each path replay was defined as the most frequently predicted trial type across all time windows in that replay. Significance tests were conducted using a Monte Carlo method with two random shuffling modes: cell identity and decoded location (5000 times each). The path replays with p < 0.05 for both shuffle modes were defined as episodic replays.

## Analysis software

All analyses were performed using custom-made programs based on Matlab (v8.3; MathWorks) and R (*R Development Core Team, 2013*) functions.

## Histology

After the rats were sacrificed by pentobarbital sodium overdose and perfused with formalin, their brains were cut coronally at 30 μm and stained with cresyl violet. The location of the tip of each electrode was estimated.

## Acknowledgements

I would like to thank Y Sakurai and F Fujiyama for discussion and comments on the manuscript, H Shimazaki for helpful suggestions on the prediction method. This study was supported by the SCOPE from Ministry of Internal Affairs and Communications (MIC) (152107008), by the JST PRESTO, and by the JSPS KAKENHI (24300148 and 25560435).

## Additional information

### Funding

| Funder | Grant reference | Author |
| --- | --- | --- |
| Japan Society for the Promotion of Science (JSPS) | 24300148, 25560435 | Susumu Takahashi |
| Ministry of Internal Affairs and Communications (MIC) | 152107008 | Susumu Takahashi |
| Japan Science and Technology Agency (JST) | PRESTO | Susumu Takahashi |

The funders had no role in study design, data collection and interpretation, or the decision to submit the work for publication.

### Author contributions

ST, Conception and design, Acquisition of data, Analysis and interpretation of data, Drafting or revising the article

### Ethics

Animal experimentation: All procedures were approved by the Doshisha University (approved number: 1229) and Kyoto Sangyo University (approved numbers: 2011-04 and 2011-05) Institutional Animal Care and Use Committees.

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
