## [Decision Letter]

Thank you for submitting your work entitled “Episodic-like memory trace in awake replay of hippocampal place-cell activity sequences” for peer review at *eLife*. Your submission has been evaluated as potentially interesting by Eve Marder (Senior editor) and three reviewers, one of whom is a member of our Board of Reviewing Editors.

The reviewers have discussed the reviews with one another and the Reviewing editor has drafted this decision to help you prepare a revised submission. For your interest, at the very bottom of this letter we have included the initial reviews, so that you can see where the summary conclusions come from.

Summary:

This paper reports that one can predict which actions a rat will take, from replay sequences and firing rates at a decision point. This extends previous work that shows that it is possible to predict goals from sequences of brain activity. The key differences are that the nature of the task is a non-spatial datum, and that this is encoded not in sequences but in firing rates of replays.

Essential revisions:

The reviewers all felt that the paper was potentially interesting. They also agreed that the paper was quite confusing and needs considerable rewriting and clarification of methodology. It especially needs much improved reporting of data and a more thorough statistical analysis. This must include:

1) Direct representation of neuronal recordings.

2) Other forms of analysis of the p-values of the replay sequences.

3) A distinct analysis of encoding beyond the Bayesian decoder.

4) The author must also provide evidence to support the conclusion that “replay in the junction encodes trial-type” is more than just representation of available sensory cues.

The reviewers felt they need to see these clarifications in order to make a final decision on the paper.

Reviewer #1:

This paper reports that one can predict which actions a rat will take, from replay sequences and firing rates at a decision point. There is some similarity in the overall result with David Foster's work on predicting goals. The key differences stated by the author is that task demand information is non-spatial, and that this is encoded not in sequences but in firing rates of replays.

In general I found the findings of the paper potentially quite interesting, but it turned out to be very difficult to track down the source of each of the presented conclusions. The key measurements were under-emphasized. The author seems to have put a layer of analysis all over the direct data, and everything is viewed through the filter of a Bayesian decoder. The paper is confusing in many places.

Major comments:

1) I would like to see direct data showing the results. The figures don't show rates or rasters, instead using a Bayesian decoder for almost all figures. I really think the author needs to use an alternative and more direct method to support each of the conclusions, since much is hidden by the inner workings of the decoder. For example, there is a small sample of a replay in Figure 3—figure supplement 1. I would like to see each of the points about replay events illustrated by a raster, a firing rate histogram, or other forms.

2) The paper lacks clarity in explaining where each of the conclusions comes from. For example, the author states in the Discussion that the task demand is encoded in “temporally compressed firing rates of place cells”. I struggled to find where this result was obtained in the paper, not least because firing rates were hidden from view as mentioned above. This turns out to be mentioned almost in passing in subsection “Replay is enhanced by spatial working memory demand”, and is easy to miss because it refers to the supplemental figure in Materials and Methods, rather than to presented data.

3) The paper builds a case for task demand representation in replay, and asserts that the task demand information is non-spatial. At face value the task demand can also be interpreted as a spatial datum: where to go next? I would ask that the author clarifies why this should be treated as non-spatial.

*Reviewer #2*:

The author of this study investigates the phenomenon of place cell sequence replay during awake states under several experimental conditions. There are several findings, most of which are confirmatory of previous studies. The main, new finding of the study is that replay is specific to different experimental conditions. While this finding is interesting, there are a number of important aspects that need to be first clarified in a revised version.

First, the author claims that different types of trials result in differential neuronal activity in the hippocampus. However, what is important and should be shown is how different was the neuronal activity across these types of trials and how different was the decoding of this activity. These should be shown at least with average (and standard deviation/error) probability values across conditions in addition to the assessment of overall significance, considering and comparing all experimental possibilities like it was done in Figure 1.

Second, the reported decoded probabilities are quite small in their average value, which brings into question how did they pass the significance test from control shuffles. Will the main effect still hold if a lower p-value (e.g., 0.01) is chosen as significance for each individual decoded event?

Third, what are some of the mechanisms by which the recruitment of context-specific replay is achieved in this study? Is the medial forebrain bundle stimulation administered as reward playing any role in the context-specific replay?

Finally, what could be the meaning of the replay events presented here, given that a quite large proportion of events significantly representing different and multiple experiences can occur spontaneously even in naive animals, such as in the case of pre-play?

Reviewer #3:

Previous studies have suggested the replay of hippocampal firing sequences that occur during SWR may encode past or future task related behaviors. Here the author uses a nice design to further address this important question. Rats are trained on a task-switching paradigm in which whether they turn left or right at the junction of a Figure 8 maze is determined by visual cues (visual discrimination) or from memory (alternation tasks). Importantly the rats do not know until the juncture which type of task they are performing (but see below). One of the main findings is that by analyzing the spike patterns during replay with a Bayesian decoder it was possible to predict the trial type/condition when the replay occurred at the juncture, but not in the start point or stem.

While the experiments seem well executed, my concern is that it is not seem that the conclusions are supported by the results.

The main finding that replay in the juncture encodes trial-type is difficult to interpret, because at the juncture point trial type is also present in the available sensory cues (that is, the rat can see what the trial type is based on the lights). While the author attempts to address this issue I did not find this conclusion compelling.

I don't think the statements that trial type is predicted during replay in the stem junction are backed by any statistics. While the red bars are above the black in Figure 3 only two trial types are shown (why?). Additionally, I think Figure 3 is mentioned out of order in the text.

It is stated that the rat only knows the correct response at the decision point (see Results section). But in the DA case doesn't it know it will have to alternate (as stated in subsection “Rats' behavior during brief periods of immobility”). Is DA versus NA encoded during the replay in the maze stem?

While trial prediction was well above chance, it was not clear how trial type was encoded. E.g. in Figure 1 conditions VDlr, NAlr, and DAlr look very similar, nevertheless the decoder extracts the trial type. How is this information encoded. Are some place cells only place cells during specific trial types? The relationship between Figure 1 was not clear. Figure 1 on the VDrr trials most of the predictions were wrong (sum of the black was more than the red). Why does this not show up in the confusion matrix for Rat 1?

---

## [Author Response]

*The reviewers all felt that the paper was potentially interesting. They also agreed that the paper was quite confusing and needs considerable rewriting and clarification of methodology. It especially needs much improved reporting of data and a more thorough statistical analysis*.

The entire manuscript has been thoroughly rewritten and reconstructed. Particularly, the specific corrections outlined below were made. According to the reviewers' suggestions, I have modified several paragraphs with a few examples that demonstrate the presented conclusions. To further clarify my clarification of methodology and data reporting, I added five figures and several subfigures showing direct data together with comments. I also performed five additional analyses.

*1) Direct representation of neuronal recordings*.

In the revised manuscript, I have added some spike raster plots to Figure 1 and Figure 3 to clarify the source of each of the presented conclusions. In addition to the raster plots, I have added corresponding firing rate histograms to Figure 3 to show the differences in the firing rates of neurons participating in the replay between different trial types. To explain such direct data in relation to the presented conclusions, I have added a few paragraphs to the Results section. In Figure 3, I show nine representative replays to explain the source of my conclusions.

*2) Other forms of analysis of the p-values of the replay sequences*.

Firstly, to illustrate my analysis of the p-values of the replay sequences in another way, I have shown the source of the p-values (posterior probability) of the replay sequences in Figure 5 and Figure 5—figure supplement 1. Secondly, to check whether matches between predicted and actual trial types were coincidental, I analyzed the probability of trial type prediction across all replays occurring at the junction on a time-window basis. There was a significant effect of probability of trial type prediction for five trial types (NALR, DALR, VDRL, NARL and DARL) in episodic replays (ANOVA, P < 0.01). Predicted trial types that showed the highest probability in replays matched the actual ones for NALR, VDRL, NARL and DARL trials. Post hoc comparisons using the Tukey HSD test indicated that the predicted trial types with the highest probabilities were significantly different from most of the others, suggesting that matches were not coincidental. I have added those results to Figure 7—figure supplement 3.

*3) A distinct analysis of encoding beyond the Bayesian decoder*.

As a separate analysis of encoding without using the Bayesian decoder, I applied a simple spatial reconstruction algorithm based on place fields during running to the replay events. To estimate representation in the replays, the maze place maps of the neurons participating in the replay were simply summed up according to the number of spikes. I added a detailed description of this methodology to the Materials and Methods section, and added Figure 6 to illustrate the results. Path preference estimated using the simple spatial reconstruction algorithm also showed an upcoming path to a memory-guided goal, similarly to the Bayesian decoder. In addition, I used the Bayesian decoder in the trial type prediction method, so I replaced it there as well with the simple spatial reconstruction algorithm. Similarly, the trial types predicted using the simple method matched the actual ones moderately well only at the junction. I have added these results to Figure 7—figure supplement 2.

*4) The author must also provide evidence to support the conclusion that “replay in the junction encodes trial-type” is more than just representation of available sensory cues*.

In line with the reviewers' suggestion, I have performed two additional analyses to assess the influence of available sensory cues on the representation in the replay. In the visually guided trials, only one of the visual cues was present at the junction. I then investigated whether the trial type encoded in the replay was just a representation of the available sensory cues. To this end, I examined whether the difference in the number of visual cues (one or both) could be used to distinguish VD from the other subtasks encoded in replays occurring at the junction, which was not the case (Figure 8; Cohen's κ = 0. 22, P = 0.35, n=18). I also investigated whether the barrier that appeared at the center of the stem in the DA trials could be a sensory cue that distinguishes NA from DA. However, the subtask predicted at the center of the stem did not match the actual NA and DA subtasks (Figure 8; Cohen's κ = 0. 27, P = 0.615, n=9). These results suggest that the trial type representation in the replay is more than just a representation of external cues. I have added these results to Figure 8.

Reviewer #1:

*1) I would like to see direct data showing the results. The figures don't show rates or rasters, instead using a Bayesian decoder for almost all figures. I really think the author needs to use an alternative and more direct method to support each of the conclusions, since much is hidden by the inner workings of the decoder. For example, there is a small sample of a replay in*
Figure 3—figure supplement 1*. I would like to see each of the points about replay events illustrated by a raster, a firing rate histogram, or other forms*.

In the revised manuscript, I have added some spike raster plots to Figure 1 and Figure 3 to clarify the source of each of the presented conclusions. In addition to the raster plots, I have added corresponding firing rate histograms to Figure 3 to show the differences in the firing rates of neurons participating in the replay between different trial types. To explain such direct data in relation to the presented conclusions, I have added a few paragraphs to the Results section. In Figure 3, I show nine representative replays to explain the source of my conclusions.

As a separate analysis of encoding without using the Bayesian decoder, I applied a simple spatial reconstruction algorithm based on place fields during running to the replay events. To estimate representation in the replays, the maze place maps of the neurons participating in the replay were simply summed up according to the number of spikes. I added a detailed description of this methodology to the Materials and Methods section, and added Figure 6 to illustrate the results. Path preference estimated using the simple spatial reconstruction algorithm also showed an upcoming path to a memory-guided goal, similarly to the Bayesian decoder. In addition, I used the Bayesian decoder in the trial type prediction method, so I replaced it there as well with the simple spatial reconstruction algorithm. Similarly, the trial types predicted using the simple method matched the actual ones moderately well only at the junction. I have added these results to Figure 7—figure supplement 2.

*2) The paper lacks clarity in explaining where each of the conclusions comes from. For example, the author states in the Discussion that the task demand is encoded in “temporally compressed firing rates of place cells”. I struggled to find where this result was obtained in the paper, not least because firing rates were hidden from view as mentioned above. This turns out to be mentioned almost in passing in subsection “Replay is enhanced by spatial working memory demand”, and is easy to miss because it refers to the supplemental figure in Materials and Methods, rather than to presented data*.

In line with this comment, I have added raster plots and corresponding firing rate histograms to Figure 3 to clarify that the subtask is encoded in the temporally compressed firing rates of place cells. To explain the results, I have added a paragraph to the Results section as follows.

“I observed that the firing rate of place cells participating in replays during different subtasks was dramatically different between subtasks even if the replay depicted a similar decoded path. For instance, consider the firing rate histograms for participating neurons in replays occurring in the start zone, central stem or junction (Figure 3). As expected from the timing of temporally compressed firings across place cells, the maximum firing rates of place cells during replays (∼80Hz) were about 10 times faster than typical maximum firing rates during running (∼8Hz).”

In addition, I have performed an additional analysis to estimate the temporal compression rate of place cell firing in the replays. I have added the result to Figure 3—figure supplement 3, and added some sentences to the text, explaining the results as follows.

“To estimate the exact temporal compression rate of place cell firing during replays, I divided the average interspike interval during candidate replays by that during the entire rest of the session. The median of the estimated compression rate was 9.0 (Figure 3—figure supplement 3), suggesting that similarly to the timing of firing, place cell firing rates were also temporally compressed during replays. During replays occurring at the junction, the raster plots of spiking activity in DARL and VDRL trials recorded from rat II showed a similar activity sequence (Figure 3, rightmost, arrows). The replays also depicted a similar decoded path. However, the firing rates during DARL trials were substantially greater than during VDRL trials, suggesting that the firing rates of neurons participating in the replay may contain information on the differences between subtasks.”

3) The paper builds a case for task demand representation in replay, and asserts that the task demand information is non-spatial. At face value the task demand can also be interpreted as a spatial datum: where to go next? I would ask that the author clarifies why this should be treated as non-spatial.

In line with this comment, I have added the following sentence to the Introduction: “Since the rats ran along similar spatial paths among while performing different subtasks throughout this task, the difference between subtasks can be interpreted as non-spatial ‘what’ information.”

Reviewer #2:

*First, the author claims that different types of trials result in differential neuronal activity in the hippocampus. However, what is important and should be shown is how different was the neuronal activity across these types of trials and how different was the decoding of this activity. These should be shown at least with average (and standard deviation/error) probability values across conditions in addition to the assessment of overall significance, considering and comparing all experimental possibilities like it was done in*
Figure 1.

In the revised manuscript, I have added some spike raster plots to Figure 1 and Figure 3 to clarify the source of each of the presented conclusions. In addition to the raster plots, I have added corresponding firing rate histograms to Figure 3 to show the differences in the firing rates of neurons participating in the replay between different trial types. To explain such direct data in relation to the presented conclusions, I have added a few paragraphs to the Results section. In Figure 3, I show nine representative replays to explain the source of my conclusions.

To perform an analysis for replays (Figure 1 of the previous manuscript), I analyzed the probability of trial type prediction across all replays occurring at the junction on a time-window basis. There was a significant effect of probability of trial type prediction for five trial types (NALR, DALR, VDRL, NARL and DARL) in episodic replays (ANOVA, P < 0.01). Predicted trial types that showed the highest probability in replays matched the actual ones for NALR, VDRL, NARL and DARL trials. Post hoc comparisons using Tukey HSD test indicated that the predicted trial types with the highest probabilities were significantly different from the most of others, suggesting that matches were not coincidental. I have added those results to Figure 7—figure supplement 3.

*Second, the reported decoded probabilities are quite small in their average value, which brings into question how did they pass the significance test from control shuffles*. *Will the main effect still hold if a lower p-value (e.g., 0.01) is chosen as significance for each individual decoded event?*

The criteria to identify statistically significant replays were defined as a continuous path showing the highest a posteriori probability with sufficient length and duration. The decoded probability per se was thus not considered for the significance test from control shuffles. In line with this comment, I have added locations showing the highest a posteriori probability in the Figure 5—figure supplement 1.

If a lower p-value is chosen, the majority of the main effects still hold, but a few of these (e.g. erroneous trials) could not be assessed because of the small sample sizes of replay events.

*Third*, *what are some of the mechanisms by which the recruitment of context-specific replay is achieved in this study? Is the medial forebrain bundle stimulation administered as reward playing any role in the context-specific replay?*

*Finally*, *what could be the meaning of the replay events presented here, given that a quite large proportion of events significantly representing different and multiple experiences can occur spontaneously even in naive animals, such as in the case of pre-play?*

In line with above comments #3 and #4, I have rewritten a paragraph in the Discussion section as follows.

“The sequential activation of place cell activity associated with SWRs can also represent paths where the animal has never experienced (9; 11; 10). As a result, a fairly large proportion of events representing different and multiple experiences can spontaneously occur, even in naïve animals. I speculate that the reason why paths and subtasks were encoded in temporally compressed firing timings and rates, respectively, in the replays may be to arrange such multiple preplayed episodes into a single replay event to easily recall and imagine an upcoming episode based on prior experience. In the present study, the rats were sequentially trained in the VD, NA and DA subtasks. Together with the preplay mechanism, this training protocol might have helped to recruit a consistent path representation across the three subtasks. Until now, previous studies investigating the content of preplays have primarily focused on the geography and timeline of the animals' running. Non-spatial information in the preplay may provide further insight into memory encoding and retrieval.”

Reviewer #3:

*The main finding that replay in the juncture encodes trial-type is difficult to interpret, because at the juncture point trial type is also present in the available sensory cues (that is, the rat can see what the trial type is based on the lights). While the author attempts to address this issue I did not find this conclusion compelling*.

In line with the reviewers' suggestion, I have performed two additional analyses to assess the influence of available sensory cues on the representation in the replay. In the visually guided trials, only one of the visual cues was present at the junction. I then investigated whether the trial type encoded in the replay was just a representation of the available sensory cues. To this end, I examined whether the difference in the number of visual cues (one or both) could be used to distinguish VD from the other subtasks encoded in replays occurring at the junction, which was not the case (Figure 8; Cohen's κ = 0. 22, P = 0.35, n =18). I also investigated whether the barrier that appeared at the center of the stem in the DA trials could be a sensory cue that distinguishes NA from DA. However, the subtask predicted at the center of the stem did not match the actual NA and DA subtasks (Figure 8; Cohen's κ = 0. 27, P = 0.615, n=9). These results suggest that the trial type representation in the replay is more than just a representation of external cues. I have added these results to Figure 8.

*I don't think the statements that trial type is predicted during replay in the stem junction are backed by any statistics. While the red bars are above the black in*
Figure 3
*only two trial types are shown (why?). Additionally, I think*
Figure 3
*is mentioned out of order in the text*.

To check whether matches between predicted and actual trial types were coincidental, I analyzed the probability of trial type prediction across all replays occurring at the junction on a time-window basis. There was a significant effect of probability of trial type prediction for five trial types (NALR, DALR, VDRL, NARL and DARL) in episodic replays (ANOVA, P < 0.01). Predicted trial types that showed the highest probability in replays matched the actual ones for NALR, VDRL, NARL and DARL trials. Post hoc comparisons using the Tukey HSD test indicated that the predicted trial types with the highest probabilities were significantly different from most of the others, suggesting that matches were not coincidental. I have added those results to Figure 7—figure supplement 3.

I have added three other representative replays to Figure 3 including replays occurring during three subtasks (VD, NA, DA) in the start zones, central stem and junction. I have added sentences explaining this figure to the Results section.

It is stated that the rat only knows the correct response at the decision point (see Results section). But in the DA case doesn't it know it will have to alternate (as stated in subsection “Rats' behavior during brief periods of immobility”). Is DA versus NA encoded during the replay in the maze stem?

In line with this comment, I investigated whether the barrier that appeared at the center of the stem in the DA trials could be a sensory cue that distinguishes NA from DA. However, the subtask predicted at the center of the stem did not match the actual NA and DA subtasks (Figure 8; Cohen's κ = 0. 27, P = 0.615, n = 9). These results suggest that the trial type representation in the replay is more than just a representation of external cues. I have added these results to Figure 8.

*While trial prediction was well above chance, it was not clear how trial type was encoded. E.g. in*
Figure 1
*conditions VDlr, NAlr, and DAlr look very similar, nevertheless the decoder extracts the trial type. How is this information encoded. Are some place cells only place cells during specific trial types? The relationship between*
Figure 1
*was not clear.*
Figure 1
*on the VDrr trials most of the predictions were wrong (sum of the black was more than the red). Why does this not show up in the confusion matrix for Rat 1?*

In line with this comment, I have added a few sentences in the Results section as follows.

“Since similar paths could be decoded from the place cell activity sequences during trial types with the same journey but different subtasks (e.g. VDLR, NALR and DALR), the Bayesian decoder cannot per se identify trial types. As my previous study reported (36), differences between subtasks are encoded in firing rates across place cells. Therefore, using a prediction method based on these firing rates (1), I predicted trial type from the firing rate pattern across place cells in conjunction with the decoded path. The bottom portion of Figure 1 shows trial types predicted from rat I in a single trial. In the visually guided VDRR and VDLL trials, some mismatches occurred because the rats could not accurately predict the trial type until they reached the junction. The predictions were most accurate for the spatial alternation trials. Confusion matrices between predicted and actual trial types for each rat during the entire session (Figure 1) show that a similar pattern was observed for all rats.”

As the confusion matrix for Rat I was normalized across 142 trials, such prediction errors in the VDRR and VDLL trials were not obvious. As the reviewer pointed out, the prediction errors of the VDRR and VDLL were relatively higher than for other trial types, even for Rat I in Figure 1.